# GENERATIVE SEMANTIC COMMUNICATION: DIFFUSION MODELS BEYOND BIT RECOVERY

## ABSTRACT

Semantic communication is expected to be one of the cores of next-generation AI-based communications. One of the possibilities offered by semantic communication is the capability to regenerate, at the destination side, images or videos semantically equivalent to the transmitted ones, without necessarily recovering the transmitted sequence of bits. The current solutions still lack the ability to build complex scenes from the received partial information. Clearly, there is an unmet need to balance the effectiveness of generation methods and the complexity of the transmitted information, possibly taking into account the goal of communication. In this paper, we aim to bridge this gap by proposing a novel generative diffusion-guided framework for semantic communication that leverages the strong abilities of diffusion models in synthesizing multimedia content while preserving semantic features. Concurrently, we propose a novel strategy to make diffusion models resilient to corrupted conditioning data, avoiding that heavily noise-affected conditioning may mislead the generation process. We reduce bandwidth usage by sending highly-compressed semantic information only. Then, the diffusion model learns to synthesize semantic-consistent scenes from such semantic information. We prove, through an in-depth assessment of multiple scenarios, that our method outperforms existing solutions in generating high-quality images with preserved semantic information even in cases where the received conditioning content is significantly degraded. More specifically, our results show that objects, locations, and depths are still recognizable even in the presence of extremely noisy conditions of the communication channel.

## 1 INTRODUCTION

The next sixth generation (6G) of wireless networks is expected to bring a radical change in thinking and developing communication systems (Luo et al., 2022). One promising aspect of semantic communication lies in its potential to reconstruct content that is semantically equivalent to the transmitted one, without necessarily requiring the recovery of the bits used to encode that content. This change of perspective may allow the receiver to quickly make the proper decisions directly linked to its goal, even though the bits of the received message are corrupted by any channel adverse condition(Dai et al., 2021). Recovering the right transmitted content can also be directly linked to the goal of communication. Consider the explanatory scenario of a vehicle transmitting visual information about the street. The crucial information is to correctly detect the presence of pedestrians, their positions, and their distance, which is the semantic knowledge, rather than recovering bits and carrying out a pixel-wise reconstruction of the colors or of the surrounding buildings.

Nevertheless, the existing solutions in the field continue to grapple with various challenges, thereby hindering their ability to deliver optimal results. One prominent issue is the lack of capacity to build complex scenes from the received information that may be corrupted or incomplete. Existing methods often rely on small-scale networks with limited expressiveness and are therefore limited to a few scenarios. As a consequence, the potential applications of these methods are curtailed, preventing their widespread adoption in real-world situations where the complexity and variability of data are considerable.

A novel communication paradigm capable of preserving semantic information can be developed by exploiting the potential of deep generative models. Recently, denoising diffusion probabilistic models

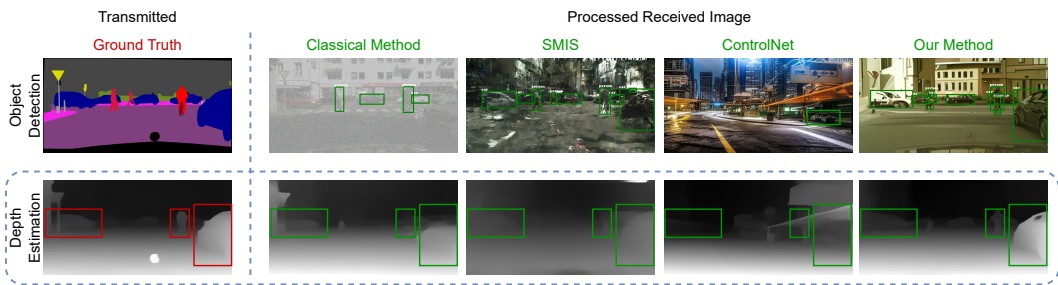

Figure 1: Synthesized images from the transmitted semantics with PSNR = 10 for the classical method, SMIS, ControlNet, and our method. The detector can still recognize objects in our generated sample, while other images are too noisy or without preserved semantics such as in ControlNet. The depth estimation confirms the better quality of our generation by correctly estimating distances from objects while producing blurred maps for comparisons.

(DDPMs) (Ho et al., 2020) have exhibited remarkable achievements in a plethora of real-world generation tasks (Saharia et al., 2022; Rombach et al., 2021; Ghosal et al., 2023; Hong et al., 2023). Among such significant results, diffusion models are able to produce photorealistic images preserving the semantic layout (Wang et al., 2022; Xue et al., 2023) in the so-called semantic image synthesis (SIS) task. The success of these models in countless domains, and especially in the SIS task, inspired us to involve them in semantic communication. However, such models usually involve clean data without corruption, and their extension to corrupted conditioning, as in the case of the communication channel, has not been investigated yet. This issue may prevent the effective deployment of diffusion models for communication purposes.

In this paper, we take a step towards bridging semantic communication and state-of-the-art generative models by presenting a novel generative semantic communication framework that meets the need for powerful models in semantic communication methods and for more robustness to corrupted conditioning in generative modeling. The core of our framework is a robust semantic diffusion model that generates photorealistic images preserving the transmitted layout. The sender transmits the compressed semantic layout over the noisy channel. The receiver collects the corrupted information and applies fast denoising to the maps before involving them in the generative process. We make the whole framework robust to bad channel conditions, ensuring that even in the case of extremely degraded received conditioning information, objects, their positions, and their depths are still recognizable in synthesized images, differently from existing approaches or from large-scale pretrained generative models. Indeed, we present a novel strategy to make the proposed semantic diffusion model robust to any noise corruption in the conditioning data, opening the path to novel resilient generative models. Moreover, our framework can significantly compress the transmitted content without causing any information loss due to the transmission of binary maps. Through a detailed assessment of seven different channel conditions and two datasets, we demonstrate the ability of our framework to generate photorealistic images consistent with the transmitted semantic information even in the case of extremely corrupted received layouts. Furthermore, we show how the proposed method allows a substantial reduction of the transmitted data rate, as it requires the transmission of binary maps only.

## 2 RELATED WORKS

Semantic communication is expected to play a key role in 6G networks (Calvanese Strinati & Barbarossa, 2020; Luo et al., 2022; Huang et al., 2023a; Qin et al., 2021). The core idea of this field is to focus on the meaning of the transmitted message, rather than on the full bit recovery. Indeed, bits may be directly affected by bad channel conditions, while the semantics may be preserved even in the case of errors at the symbolic level. This novel view of wireless communication is influencing several applications ranging from image (Patwa et al., 2020; Wang et al., 2019), to video compression and transmission (Jiang et al., 2022; AL-Shakarji et al., 2019), and it is expected to increase its impact in much more fields in the next years (Dai et al., 2021).

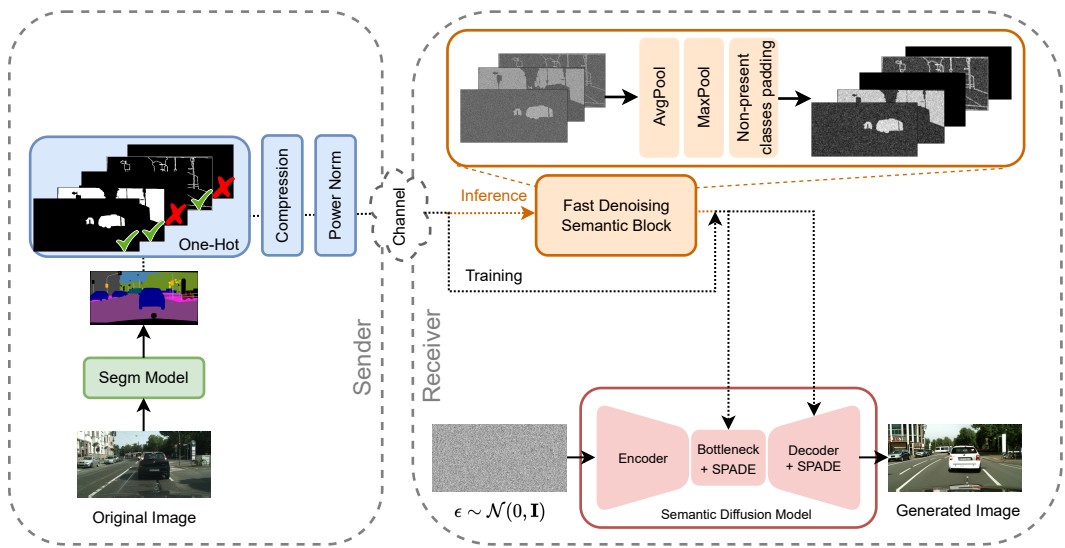

Figure 2: Proposed generative semantic communication framework. The sender transmits one-hot, compressed, and normalized encoded maps over the noisy channel. The receiver takes the noisy maps and directly involves them to train the semantic diffusion model. During inference, the receiver applies a fast denoising to the semantic information in order to improve image quality.

Diffusion models have brought a real breakthrough in generative modeling, showing impressive results in several generation tasks, ranging from image (Nichol et al., 2021; Saharia et al., 2022; Rombach et al., 2021; You et al., 2023) to audio (Ghosal et al., 2023; Popov et al., 2023; Huang et al., 2023b; Turetzky et al., 2022) or video generation (Hong et al., 2023; Singer et al., 2022; Gu et al., 2023; Jiang et al., 2023). Diffusion models synthesize samples starting from a standard Gaussian distribution and by performing an iterative denoising process up to the desired new content. This process makes diffusion model generation far stabler than generative adversarial networks (Croitoru et al., 2023). Among the tasks in which diffusion models stand out, there is semantic image synthesis (SIS), which is the task of generating images consistent with a given semantic layout. Although most SIS approaches are based on generative adversarial networks (Tan et al., 2021; Park et al., 2019; Schönfeld et al., 2021; Zhu et al., 2020; Liu et al., 2019), in the last year, a novel SIS model outperforms other approaches by involving a diffusion model for synthesizing semantically-consistent high-quality scenes (Wang et al., 2022).

Very recently, generative semantic communication methods have been introduced (Barbarossa et al., 2023). Among them, generative adversarial networks have been the first generative tool to be involved in tasks such as image compression or denoising (Han et al., 2022a; Erdemir et al., 2022). Overall, existing generative communication frameworks are often limited to quite-simple models such as small VAEs (Malur Saidutta et al., 2020; Estiri et al., 2020) or pretrained GAN generators (Erdemir et al., 2022). In addition, normalizing flows have started to be involved in semantic communications to increase framework expressiveness (Han et al., 2022b). However, these networks have been often involved in tasks that underestimate their capabilities, limiting their effectiveness.

## 3 PROPOSED METHOD

In this paper, we present a novel generative semantic communication framework based on denoising diffusion probabilistic models (DDPMs) for synthesizing high-quality images that preserve the transmitted semantic information.

### 3.1 PROBLEM SETTING

Each communication method has to face the physical challenges imposed by real-world systems. First of all, the transmitter has to respect a power constraint on the transmitted signal $\mathbf{z}$ in order to account

for the limited transmit power of the sender device. This implies $1/k \, \mathbb{E}_{\mathbf{z}}[\|\mathbf{z}\|_2^2] \leq P$ (Erdemir et al., 2022). Then, the signal flows over a noisy channel. To be consistent with the literature (Erdemir et al., 2022; Shao et al., 2021), we consider the benchmark situation of additive white Gaussian noise (AWGN), where the noise $\epsilon$ is sampled from $\epsilon \sim \mathcal{N}(\mathbf{0}, \sigma^2\mathbf{I})$, whereby $\sigma^2$ is the noise variance, and then added to the transmitted symbols as $\tilde{\mathbf{z}} = \mathbf{z} + \epsilon$. According to this formulation, the peak signal-to-noise ratio (PSNR) can be defined as

$$\text{PSNR} = 10 \log \frac{P}{\sigma^2} \, (\text{dB}). \tag{1}$$

In our simulations, we assume that the PSNR ranges from 0 to 100. For PSNR values close to 100 the channel has perfect conditions and the information flows without corruption, while for very low PSNR values (PSNR $\leq 20$) the noise can heavily corrupt the transmitted information, severely distorting the received content. Additionally, the communication channel bandwidth is usually limited and systems try to save as much bandwidth as possible to avoid bit loss or channel overload. Consequently, the transmitted information has to be significantly compressed, sometimes causing information loss. In this paper, we present a solution to address both of these communication problems, building a framework robust to any AWGN channel condition, and transmitting only extremely compressed content with negligible loss of information. We also test different channel noise such as Poisson and a mixture of noises, for which we report the analysis in Appendix D.

## 3.2 Generative Semantic Communication Framework

We introduce a novel generative semantic communication framework, whose core is a semantic diffusion model. Such a model generates high-quality images under the guidance of the semantic information brought by semantic maps. Figure 2 presents the proposed architecture including both the sender and receiver sides.

**Sender.** To process the original image and generate a semantic map, any existing segmentation model can be used, as there is no need for communication between sender and receiver networks. Previous studies have demonstrated that conditioning a model with a one-hot encoded map yields better results compared to conditioning with a single segmentation map. For this reason, we adopt such an approach. However, transmitting the one-hot encoded information over a noisy channel may cause some issues. Indeed, empty contents corresponding to non-present classes become highly corrupted data, introducing significant noise into the resulting generated image. Furthermore, transmitting such content occupies valuable channel bandwidth with irrelevant information, further deteriorating the channel conditions.

To address these problems, we propose a solution that transmits only the most informative content. By doing so, we eliminate noisy-only information while maximizing the utilization of the available bandwidth. Once this procedure is complete, we apply a very strong compression of the encoded maps. This reduces the number of bits to be transmitted without sacrificing relevant information, especially since, contrary to RGB images, the black-and-white regions of one-hot maps are minimally affected by strong compression methods, as we show in the supplementary material.

**Receiver.** The core of the proposed method lies on the receiver side. The received one-hot maps are significantly corrupted by the communication channel making them extremely noisy. Conditioning the diffusion model with such noisy content may instead inject undesirable noise in generated images or mislead the sampling process. To avoid this issue and to make the diffusion model robust to such distorted content, we train it with noisy maps and let the network weights adapt to different channel conditions. Once the model is trained, to improve the quality of generated images during inference, we also insert a fast-denoising semantic (FDS) block, whose scope is to attenuate the random noisy condition of received maps. It is important to note that, differently from previous methods (Shao et al., 2021; Erdemir et al., 2022), our receiver does not need to be aware of the channel conditions and it may work with any channel condition. Then, we sample $\mathbf{x}_0 \sim \mathcal{N}(\mathbf{0}, \mathbf{I})$ and we progressively remove noise up to synthesizing a new sample whose semantics reflects the conditioning one.

**Fast Denoising Semantic Block.** The fast-denoising semantic (FDS) block takes in input the heavily-corrupted one-hot encoded maps $\hat{\mathbf{y}}$ that have been transmitted over the noisy channel. FDS applies a fast denoise taking into account the black-and-white nature of the information. In detail, FDS produces the complete denoised semantic maps $\mathbf{y}$ by:

$$\mathbf{y} = \text{Pad}\left(\text{MaxPool}\left(\text{AvgPool}\left(\hat{\mathbf{y}}\right)\right)\right). \tag{2}$$

First, the average pooling removes noise spikes in the maps. Then, since the maps comprise large $0/1$ regions, where 1 corresponds to areas where the class is present and 0 to empty spaces, the MaxPool performs a high-pass filter operation mainly keeping the 1s regions only and discarding other values. Finally, FDS pads the clean missing classes that have been removed on the sender side.

### 3.3 SEMANTIC DIFFUSION MODEL

The core of our generative semantic communication framework is the semantic diffusion model that generates images by preserving the transmitted semantic information.

**Conditional diffusion model.** Given a sample $\mathbf{x}_0$ and a conditioning $\mathbf{y}$, the conditional data distribution follows $q(\mathbf{x}_0|\mathbf{y})$. In this setup, conditional diffusion models maximize the likelihood $p_\theta(\mathbf{x}_0|\mathbf{y})$. The reverse process is a Markov chain with learned Gaussian transitions that starts at $p(\mathbf{x}_T) \sim \mathcal{N}(0, \mathbf{I})$ and is defined as

$$p_\theta(\mathbf{x}_{0:T}|\mathbf{y}) = p(\mathbf{x}_T) \prod_{t=1}^{T} p_\theta(\mathbf{x}_{t-1}|\mathbf{x}_t, \mathbf{y}), \tag{3}$$

with $p_\theta(\mathbf{x}_{t-1}|\mathbf{x}_t, \mathbf{y}) = \mathcal{N}(\mathbf{x}_{t-1}; \mu(\mathbf{x}_t, \mathbf{y}, t), \sigma_\theta(\mathbf{x}_t, \mathbf{y}, t))$. The forward process $q(\mathbf{x}_{1:T}|\mathbf{x}_0)$ injects Gaussian noise into data following the defined variance schedule $\beta_1, ..., \beta_T$. Considering that $\alpha_t := \prod_{s=1}^{t}(1 - \beta_s)$, the forward process is defined by

$$q(\mathbf{x}_t|\mathbf{x}_0) = \mathcal{N}(\mathbf{x}_t; \sqrt{\alpha_t}\mathbf{x}_0, (1 - \alpha_t)\mathbf{I}). \tag{4}$$

**Encoder.** The U-Net (Ronneberger et al., 2015) encoder comprises an input convolution and a stack of encoder blocks with downsampling. The encoder block interleaves a convolution layer, a SiLU activation (Ramachandran et al., 2017), and a group normalization (Wu & He, 2018). The block also implements a fully-connected layer with weights $\mathbf{w}$ and bias $\mathbf{b}$ to inject the time information $t$ by scaling and shifting the mid-activation $\mathbf{a}$ by $\mathbf{a}_{i+1} = \mathbf{w}(t) \cdot \mathbf{a}_i + \mathbf{b}(t)$. Furthermore, at resolutions $32 \times 32$, $16 \times 16$, and $8 \times 8$ the encoder involves attention modules with skip connection. Given $\mathbf{x}$ input and $\mathbf{y}$ output of the attention block, and four $1 \times 1$ convolutions with weights $\mathbf{w}_f, \mathbf{w}_g, \mathbf{w}_h$, and $\mathbf{w}_v$, we define $f(\mathbf{x}) = \mathbf{w}_f\mathbf{x}, g(\mathbf{x}) = \mathbf{w}_g\mathbf{x}$ and $h(\mathbf{x}) = \mathbf{w}_h\mathbf{x}$, arriving to

$$\mathcal{M}(u, v) = \frac{f(\mathbf{x}_u)^\top g(\mathbf{x}_v)}{\|f(\mathbf{x}_u)\|\|g(\mathbf{x}_v\|}, \tag{5}$$

$$\mathbf{y}_u = \mathbf{x}_u + \mathbf{w}_v \sum_v \text{softmax}_v(\alpha\mathcal{M}(u, v)) \cdot h(\mathbf{x}_v), \tag{6}$$

whereby the spatial dimension indexes are $u \in [1, H], v \in [1, W]$.

**Decoder.** The decoder blocks are crucial for the semantic conditioning of the whole model. Indeed, to fully exploit the semantic information, decoder blocks implement spatially-adaptive normalization (SPADE) (Park et al., 2019) that replaces group normalization in the encoder. The SPADE module introduces semantic content in the data flow by adjusting the activations $\mathbf{a}_i$ as follows

$$\mathbf{a}_{i+1} = \gamma_i(\mathbf{x}) \cdot \text{Norm}(\mathbf{a}_i) + \mathbf{b}_i(\mathbf{x}), \tag{7}$$

in which $\text{Norm}(\cdot)$ is the group normalization, and $\gamma_i, \mathbf{b}_i$ are the spatially-adaptive weights and biases learned from the conditioning semantic map. It is worth noting that we train our semantic diffusion model directly with noisy semantic maps to let the network be robust to different channel noises. At each step, we simulate varying channel conditions by sampling the noise variance in $\{0.9, 0.6, 0.36, 0.22, 0.13, 0.05, 0.00\}$ corresponding to PSNRs in $\{1, 5, 10, 15, 20, 30, 100\}$, weighting perfect channel conditions (PSNR= 100) higher. We note that incorporating channel noise during training heavily impacts the quality of generated images in inference.

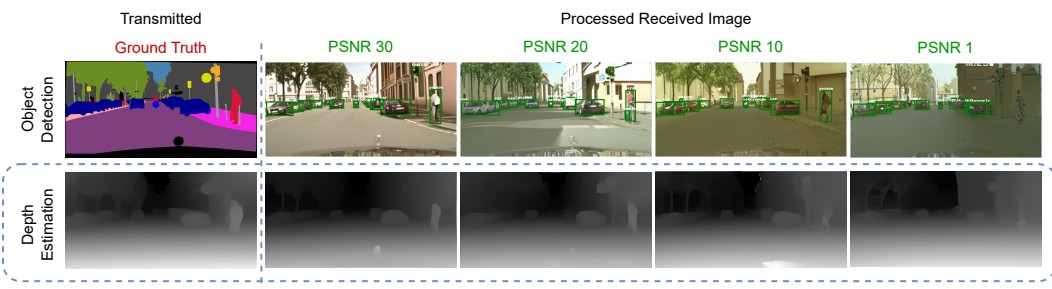

Figure 3: Our method results for different PSNR values of the communication channel. The detector recognizes well cars and pedestrians in all the samples, proving that our method works properly. Moreover, the depth estimation is consistent across all the scenarios, further validating the effectiveness of the proposed method.

## 3.4 LOSS FUNCTIONS

We train the semantic diffusion model with a combination of two loss functions. Considering an input image $\mathbf{x}$ and the sequence of time steps $t \in \{0, ..., T\}$, the corresponding noisy image $\tilde{\mathbf{x}}$ at time $t$ is built by $\tilde{\mathbf{x}} = \sqrt{\alpha_t}\mathbf{x} + \sqrt{1 - \alpha_t}\epsilon$. The noise is sampled from $\epsilon \sim \mathcal{N}(\mathbf{0}, \mathbf{I})$ and the $\alpha_t$ is the noise scheduler at $t$, where the maximum timestep is $T = 1000$. The model tries to predict the noise $\epsilon$ to reconstruct the reference image $\mathbf{x}$ according to the guidance of the semantic map $\mathbf{y}$. The denoising loss function $\mathcal{L}_\mathrm{d}$ takes the form of

$$\mathcal{L}_\mathrm{d} = \mathbb{E}_{t,\mathbf{x},\epsilon} \left[ \left\| \epsilon - \epsilon_\theta \left( \sqrt{\alpha_t}\mathbf{x} + \sqrt{1 - \alpha_t}\epsilon, \mathbf{y}, t \right) \right\|_2 \right]. \tag{8}$$

In order to improve the generated images log-likelihood, the model is trained to predict variances too Nichol & Dhariwal (2021) by means of the KL divergence between the predicted distribution $p_\theta(\mathbf{x}_{t-1}|\mathbf{x}_t, \mathbf{y})$ and the diffusion process posterior $q(\mathbf{x}_{t-1}|\mathbf{x}_t, \mathbf{x}_0)$:

$$\mathcal{L}_\mathrm{KL} = \mathrm{KL}(p_\theta(\mathbf{x}_{t-1}|\mathbf{x}_t, \mathbf{y}) \| q(\mathbf{x}_{t-1}|\mathbf{x}_t, \mathbf{x}_0)). \tag{9}$$

The resulting loss function is balanced by $\lambda$ as:

$$\mathcal{L} = \mathcal{L}_\mathrm{d} + \lambda \mathcal{L}_\mathrm{KL}. \tag{10}$$

## 3.5 CLASSIFIER-FREE GUIDANCE

The image quality of conditional diffusion models can be improved through the gradient of the log-probability distribution $\nabla_{\mathbf{x}_t} \log p(\mathbf{y}|\mathbf{x}_t)$ by perturbing the mean with a guidance-scale hyperparameter $s$ (Dhariwal & Nichol, 2021). While previous diffusion models involved a classifier for this procedure (Dhariwal & Nichol, 2021), novel methods directly leverage the generative model power to provide the gradient during the sampling step (Ho & Salimans, 2021). In our framework, we can disentangle the conditional noise estimation from the unconditional one, by involving the semantic map for the first estimate as $\epsilon_\theta(\mathbf{x}_t|\mathbf{y})$ and the null label for the second one, that is $\epsilon_\theta(\mathbf{x}_t|\mathbf{0})$ (Wang et al., 2022). The gradient of the log-probability distribution is then proportional to the difference between the estimates as

$$\epsilon_\theta(\mathbf{x}_t|\mathbf{y}) - \epsilon_\theta(\mathbf{x}_t|\mathbf{0}) \propto \nabla_{\mathbf{x}_t} \log p(\mathbf{x}_t|\mathbf{y}) - \nabla_{\mathbf{x}_t} \log p(\mathbf{x}_t) \tag{11}$$

$$\propto \nabla_{\mathbf{x}_t} \log p(\mathbf{y}|\mathbf{x}_t). \tag{12}$$

Accordingly, the noise estimation is performed by means of the disentangled component as

$$\hat{\epsilon}_\theta(\mathbf{x}_t|\mathbf{y}) = \epsilon_\theta(\mathbf{x}_t|\mathbf{y}) + s \cdot (\epsilon_\theta(\mathbf{x}_t|\mathbf{y}) - \epsilon_\theta(\mathbf{x}_t|\mathbf{0})). \tag{13}$$

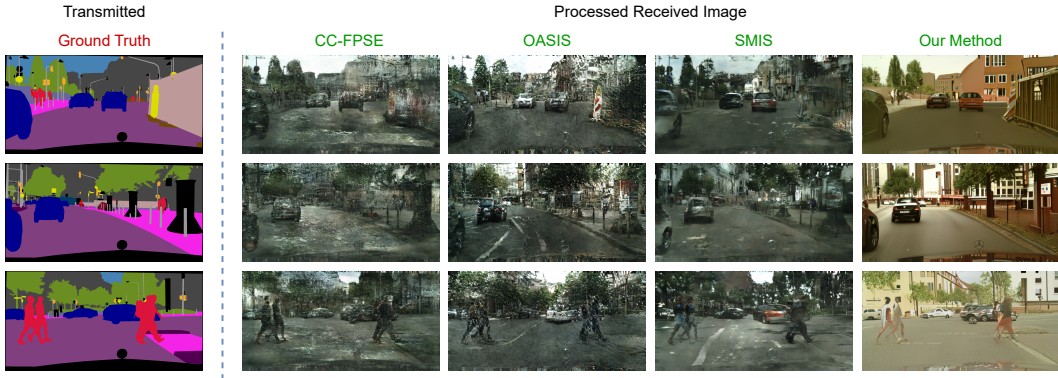

Figure 4: Comparisons among most performing models (CC-FPSE (Liu et al., 2019), OASIS (Schönfeld et al., 2021), and SMIS (Zhu et al., 2020)) with PSNR = 15. Other methods produce almost noise-only images. Our method produces the best quality samples in which it is easy to recognize objects, cars, and pedestrians, while comparisons generate scenes heavily corrupted by noise.

Table 1: Semantic evaluation of generated images under different channel conditions.

| Method | mIoU↑ | | | | | | |
|---|---|---|---|---|---|---|---|
| PSNR | 100 | 30 | 20 | 15 | 10 | 5 | 1 |
| Full image | - | **0.955**±.032 | 0.911±.155 | 0.906±.247 | 0.906±.339 | 0.240±.193 | 0.110±.298 |
| SPADE (Park et al., 2019) | 0.909±.127 | 0.914±.255 | 0.921±.315 | 0.812±.364 | 0.672±.321 | 0.253±.288 | 0.313±.144 |
| CC-FPSE (Liu et al., 2019) | 0.908±.045 | 0.908±.121 | 0.911±.315 | 0.928±.345 | 0.852±.245 | 0.653±.183 | 0.322±.284 |
| SMIS (Zhu et al., 2020) | 0.909±.064 | 0.919±.066 | 0.909±.214 | 0.931±.208 | 0.901±.244 | 0.899±.290 | 0.876±.211 |
| OASIS (Schönfeld et al., 2021) | 0.910±.111 | 0.908±.191 | 0.912±.232 | 0.697±.165 | 0.662±.356 | 0.345±.112 | 0.232±.191 |
| SDM (Wang et al., 2022) | 0.921±.051 | 0.340±.022 | 0.333±.061 | 0.351±.011 | 0.297±.021 | 0.256±.019 | 0.211±.043 |
| Our method | **0.940**±.014 | 0.942±.212 | **0.944**±.297 | **0.945**±.141 | **0.905**±.112 | **0.913**±.214 | **0.925**±.111 |

# 4 EXPERIMENTAL EVALUATION

In this Section, we report the experimental setup and the results of the tests we conduct.

## 4.1 SETUP

**Datasets.** We involve Cityscapes, which contains 35 classes, and COCO-Stuff, with 183 classes, as our datasets for training and evaluation. Both these datasets comprise instance annotations that we consider in our framework.

**Evaluation.** The purpose of a semantic communication framework is to properly transmit the meaning of the image the sender wants to communicate to the receiver. Therefore, the crucial part of the evaluation is measuring the preserved semantic meaning in the synthesized images from the receiver. To this end, in addition to image quality evaluation with FID and LPIPS, we perform three different types of assessment. First, we compute and objectively evaluate the semantic interpretability of the generated images by building the semantic maps of the latter and comparing them with the original ones. We compute the mIoU metric on segmentation maps of generated images obtained through a pretrained model. For this evaluation, we employ DRN-D-105 (Yu et al., 2017) on Cityscapes, and MaskFormer (Cheng et al., 2021) on COCO-Stuff. Note that the mIoU evaluation strongly depends on the effectiveness of the pretrained model involved to compute the segmentation maps. Second, we evaluate how much the synthesized images preserve objects meaning which is crucial for semantic communication systems in autonomous driving. As an example, buildings or landscapes can be badly generated as long as pedestrians or bicycles are well-recognized by the car. For this evaluation, we employ DETR (Carion et al., 2020). Third, another key aspect of autonomous driving is depth estimation, which helps estimate the distance between objects (Fonder et al., 2021), thus we evaluate this aspect via DPT (Ranftl et al., 2021).

Table 2: Perceptual similarity evaluation of generated images under different channel conditions.

| Method | LPIPS↓ | | | | | | |
|---|---|---|---|---|---|---|---|
| PSNR | 100 | 30 | 20 | 15 | 10 | 5 | 1 |
| Full image | - | 0.623±.074 | 0.684±.165 | 0.713±.054 | 0.730±.156 | 0.747±.154 | 0.738±.186 |
| SPADE (Park et al., 2019) | **0.546**±.045 | 0.565±.072 | 0.603±.022 | 0.726±.019 | 0.792±.115 | 0.824±.054 | 0.827±.011 |
| CC-FPSE (Liu et al., 2019) | **0.546**±.025 | 0.559±.004 | 0.581±.009 | 0.620±.011 | 0.855±.024 | 0.753±.032 | 0.812±.055 |
| SMIS (Zhu et al., 2020) | **0.546**±.002 | 0.548±.030 | 0.561±.010 | 0.574±.021 | 0.603±.027 | 0.649±.044 | 0.680±.124 |
| OASIS (Schönfeld et al., 2021) | 0.561±.032 | 0.564±.054 | 0.580±.012 | 0.613±.073 | 0.679±.020 | 0.783±.034 | 0.828±.122 |
| SDM (Wang et al., 2022) | 0.549±.061 | 0.543±.072 | 0.555±.066 | 0.599±.043 | 0.606±.071 | 0.655±.098 | 0.749±.119 |
| Our method | 0.606±.032 | **0.517**±.004 | **0.523**±.011 | **0.542**±.003 | **0.549**±.009 | **0.620**±.023 | **0.609**±.042 |

Table 3: Generation quality evaluation of generated images under different channel conditions.

| Method | FID×10 ↓ | | | | | | |
|---|---|---|---|---|---|---|---|
| PSNR | 100 | 30 | 20 | 15 | 10 | 5 | 1 |
| Full image | - | **6.284**±.053 | 13.684±.032 | 20.045±.865 | 28.005±.878 | 37.931±.639 | 42.004±.911 |
| SPADE (Park et al., 2019) | 10.324±.171 | 14.200±.179 | 22.971±.190 | 42.681±.201 | 55.420±1.056 | noise | noise |
| CC-FPSE (Liu et al., 2019) | 24.590±.056 | 20.337±.060 | 26.253±.049 | 33.166±.210 | 40.374±.345 | noise | noise |
| SMIS (Zhu et al., 2020) | **8.758**±.162 | 9.147±.100 | **11.750**±.095 | 14.775±.129 | 21.373±.167 | 34.586±.171 | 44.115±.412 |
| OASIS (Schönfeld et al., 2021) | 10.403±.053 | 10.339±.099 | 16.179±.122 | 24.892±.134 | 40.440±.349 | noise | noise |
| SDM (Wang et al., 2022) | 9.899±.391 | 16.642±2.101 | 31.510±2.926 | noise | noise | noise | noise |
| Our method | 11.848±.061 | 12.355±.090 | 14.030±.201 | **14.008**±.251 | **14.851**±.193 | **15.315**±.349 | **15.989**±.561 |

## 4.2 COMPARISONS AND RESULTS ANALYSIS

We compare our proposal with classical communication methods that directly transmit the image over the channel. For more comparisons, we consider well-known semantic image synthesis models such as SPADE (Park et al., 2019), CC-FPSE (Liu et al., 2019), SMIS (Zhu et al., 2020), OASIS (Schönfeld et al., 2021), and SDM (Wang et al., 2022), and ControlNet (Zhang & Agrawala, 2023) as we show in Fig. 1. Details on the comparison methods are available in Appendix B. We consider different channel scenarios, ranging from extremely degraded conditions to perfect transmissions, by setting PSNR values in $\{1, 5, 10, 15, 20, 30, 100\}$, and two datasets that are Cityscapes and COCO-Stuff.

**Advantages.** Based on the achieved results, we can see how the proposed method clearly outperforms its competitors according to all the metrics used in our assessment. In particular, our approach is far more robust to bad channel conditions, still preserving semantics meanings with PSNR $\leq 10$ according to the mIoU metric, as Table 1 shows for the Cityscapes dataset. Moreover, it also generates more perceptually similar samples with respect to all the other comparisons, as measured by the LPIPS metric in Table 2. Indeed, in the context of communications, the lower the LPIPS between the original image and the synthesized one, the better the generation is since the two images are perceptually similar (Han et al., 2022a). Few methods manage to synthesize meaningful images with very low PSNR values and the sample quality deteriorates as the channel conditions worsen. On the contrary, as Table 3 reports, the FID of the proposed method samples is the lowest and quite stable across all the different scenarios, proving that the generation of our model is robust to every channel condition. Figure 1 reports generated samples with PSNR $= 10$ of the classical method, SMIS, ControlNet, and our method. Our sample is of better quality and its depth is much closer to the original one with respect to comparisons that produce noisy images and blurred depths. Additionally, Figure 3 shows generated samples or different channel scenarios and the detected objects by DETR. Even in the case of PSNR $= 1$, DETR still recognizes the largest part of the objects. Furthermore, the DPT depth estimation also gives consistent results across different conditions, with synthesized images at low PSNRs preserving the depths similar to the original image depth. As a further comparison for image quality, we show diverse samples in Figure 4, where our method is compared to the three best comparisons. Although CC-FPSE, OASIS, and SMIS produce better samples with respect to other models, our method clearly generates the best quality samples. Indeed, in our samples, objects, cars, and pedestrians are clearly recognizable, whereas in other images they are blurred or noisy. On the COCO dataset, existing approaches produce noisy samples, while our method still provides meaningful samples able to achieve good performance, as Table 4 shows.

**Bit rate**. A crucial aspect for communication frameworks is saving bandwidth and reducing the number of transmitted bits. To this end, we transmit over the channel the compressed one-hot-encoded maps that crucially reduce the number of encoded bits with respect to classical communications. We conduct the evaluation on the Cityscapes dataset with images resized to $256 \times 512$. Transmitting the

Table 4: Semantic, perceptual similarity, and generation quality evaluation with fixed PSNR = 10 on the COCO-Stuff dataset.

| Method | mIoU↑ | LPIPS↓ | FID×10 ↓ |
|---|---|---|---|
| Full image | 0.331±.145 | 0.687±.003 | 40.562±2.513 |
| SPADE (Park et al., 2019) | noise | noise | noise |
| CC-FPSE (Liu et al., 2019) | noise | noise | noise |
| SDM (Wang et al., 2022) | noise | noise | noise |
| Our method | **0.365**±.096 | **0.683**±.011 | **36.664**±1.527 |

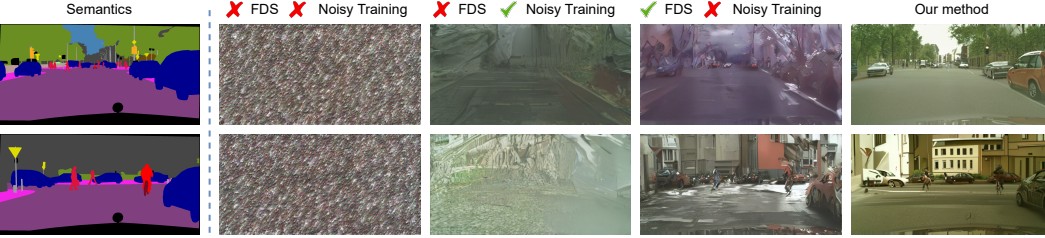

Figure 5: Generated samples from ablation studies with PSNR = 10. Samples without FDS and noisy training are clearly noisy. Then, both FDS and noisy training help improve sample quality.

full image requires 1464000 bits, while, on average, the proposed method just needs 119000 bits, with a considerable reduction of 92%, proving a further advantage of the proposed method.

**Limitations.** Although our framework achieves excellent results in every scenario we test and outperforms existing solutions, clearly there is still a lot to do to improve the proposed method before directly applying it in production-ready systems. Surely, improvements should be made on computational efficiency in terms of sampling time, energy consumption, and transmission latency. However, this complicated aspect requires a detailed study that is out of the scope of this paper that only aims at introducing an innovative robust generative semantic communication framework whose core is a powerful semantic diffusion model.

## 4.3 ABLATION STUDIES

We perform ablation tests to corroborate our method choices. We study the inference performances with and without the proposed FDS block and without the noisy maps during training, fixing the PSNR to 10. Table 5 shows the effectiveness and the importance of both the proposed noisy training and the FDS module in inference. Figure

Table 5: Ablation results on the Cityscapes dataset.

| FDS | Noisy training | LPIPS | mIoU |
|---|---|---|---|
| ✗ | ✗ | noise | noise |
| ✗ | ✓ | 0.665 | 0.613 |
| ✓ | ✗ | 0.663 | 0.713 |
| ✓ | ✓ | **0.549** | **0.905** |

5 allows for a visual inspection of the generated results without the proposed methods. While the semantic diffusion model alone (✗ FDS ✗ Noisy Training) produces only noise, both FDS and the noisy training help to improve samples quality.

## 5 CONCLUSION

To the best of our knowledge, this paper presents the first generative semantic communication framework whose core is a semantic diffusion model. In detail, we make the whole framework robust to bad channel conditions by training the semantic diffusion model with noisy semantics, and by inserting a fast denoising semantic block to improve inference image quality. Furthermore, we crucially reduce the amount of information to transmit by sending over the channel the present-classes binary maps only. Our performance assessment highlights that the proposed framework generates semantically-consistent samples even in the case of extremely degraded channel conditions, outperforming all other competitors.

## REPRODUCIBILITY STATEMENT

We strongly believe research and science are based on the reproducibility of existing works. Therefore, in the Experimental Details Section of the Appendix we report all the details to reproduce our experiments, including image sizes and preprocessing, all the hyperparameters and network architecture, the hyperparameters of the diffusion process and of the sampling procedure, as well as the details on the hardware we employ for the experiments. Although the code and the pretrained checkpoints will be freely accessible at the end of the revision process, we include them in a zip folder as supplementary material for the submission.

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

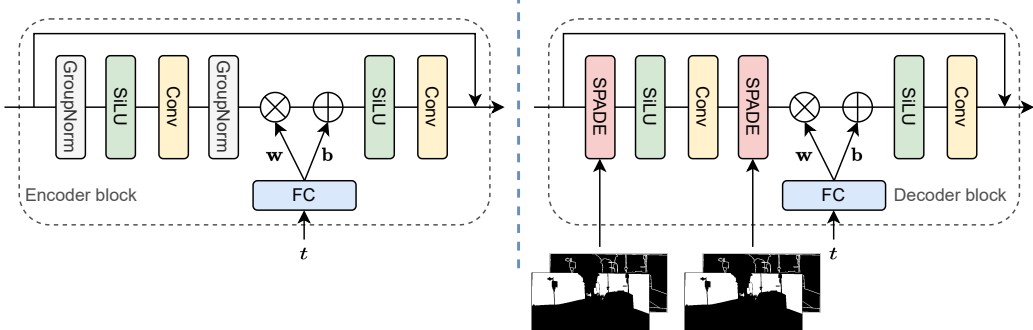

Figure 6: Encoder and decoder blocks of our U-Net-based semantic diffusion model.

## APPENDIX

This appendix includes additional details, experiments, and explanations of the proposed method.

## A  FROM TECHNICAL TO SEMANTIC COMMUNICATION CHALLENGES

Since the channel capacity formula by Shannon in 1948 (Shannon, 1948), communication systems have grown from first-generation (1G) to Beyond-fifth generation (B5G), progressively approaching the non-physical-layer capacity limit and designing new frontiers in line with users' needs. In 1953, Weaver theorized that communication challenges can be enclosed in three gradual levels (Weaver, 1953):

1. **The technical challenge.** It deals with the classical Shannon's communication theory and focuses on the proper way of transmitting bits from a sender to a receiver.

2. **The semantic challenge.** Rather than just transmitting bits, this level should account for properly transmitting the meaning of the messages the sender wants to communicate to the receiver.

3. **The effectiveness challenge.** This level deals with the efficiency of the transmission of previous levels.

With the upcoming advent of the sixth generation (6G), a radical rethinking of communication framework design has started, sliding from the first to the second level of Weaver's theory (Calvanese Strinati & Barbarossa, 2020; Luo et al., 2022). In this switch, generative learning methods are making their way bringing considerable improvements in several communication tasks, such as content compression or denoising (Han et al., 2022a; Erdemir et al., 2022). However, generative communication frameworks are often limited to quite-simple models such as small VAEs (Malur Saidutta et al., 2020; Estiri et al., 2020) or pretrained GAN generators (Erdemir et al., 2022). Moreover, these networks have been involved in tasks that underestimate their capabilities, limiting their effectiveness. On the contrary, the enormous power of recent generative models may lead to profoundly transform semantic communications.

## B  EXPERIMENTAL DETAILS

We provide additional details to reproduce our experiments. Finally, we set the guidance scale $s$ equal to 2 for Cityscapes and 2.5 for COCO-Stuff. We resize Cityscapes images to $256 \times 512$, and COCO-Stuff images to $256 \times 256$. We train the model with PyTorch on a single NVIDIA Tesla V100 GPU (32GB) for the Cityscapes and on a single NVIDIA Quadro RTX8000 (48GB) for COCO-Stuff. We use a batch size of 4 in all the experiments, a learning rate of 0.0001 for the AdamW optimizer, and attention blocks at resolutions $32, 16,$ and8 with a number of head channels equal to 64. The features dimension in the encoder and in the decoder of the U-Net model is halved at each layer, while comprising a number of channels equal to $[256, 256, 512, 512, 1024,$ and 1024]. For sampling, we

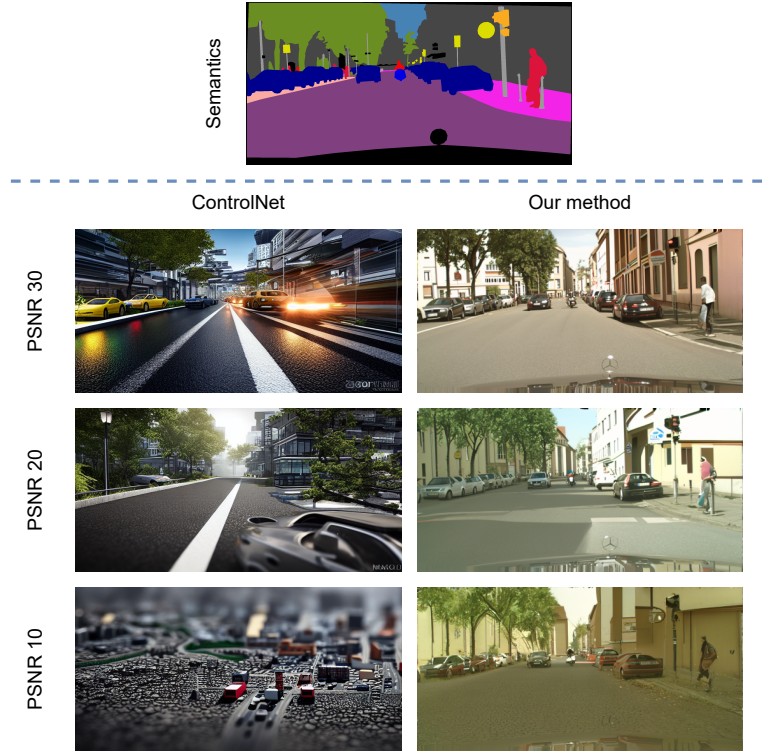

Figure 7: Generated samples under different channel conditions (PSNR in $\{30, 20, 10\}$) by ControlNet and by the proposed method. While for good channel conditions (PSNR= 30) **CnotrolNet** generates a meaningful sample, for lower values of the PSNR, therefore with more corrupted information received, it **completely loses the ability to preserve the semantics in generated samples**, making it unusable in communication systems.

set the number of diffusion steps to $T = 1000$ with a linear noise schedule. We use mixed precision for training in order to reduce the computational complexity. The loss balance term $\lambda$ is set to $0.001$, according to (Wang et al., 2022). Furthermore, we involve an exponential moving average of the U-Net network weights with a decay equal to $0.9999$. Figure 6 shows the structure of our encoder and decoder blocks. The code and the checkpoints will be freely available at the end of the revision process.

**Comparison methods details.** We compare the proposed methods with several comparison methods that include bot GAN-based and diffusion-based generative models and we add here the details about such models. For each model, we follow the hyperparameters setting of the original paper. SPADE Park et al. (2019) is a GAN in which the author proposes to insert spatially-adaptive normalization in the generator network. CC-FPSE Liu et al. (2019) is based on a GAN structure with a generator conditioned through a weight prediction network and a semantics-embedding discriminator. SMIS Zhu et al. (2020) is a GAN model, whose generator is based on a Group Decreasing Network and an encoder-decoder structure. OASIS Schönfeld et al. (2021) is a GAN-based model in which the generator is conditioned by the 3D noise and label map and the discriminator is segmentation-based. SDM Wang et al. (2022) is a recently introduced diffusion model that utilizes SPADE modules in the decoder of the U-Net structure. Finally, ControlNet Zhang & Agrawala (2023) is a recent technique to condition diffusion models by locking the original network and creating a trainable copy with conditioning that is then connected with the first one using zero convolution layers.

## C COMPARISON WITH PRETRAINED GENERATIVE MODELS

Although several recent works directly involve pretrained generative models, in a communication scenario it is hard to follow this procedure as these models have not been engineered and trained

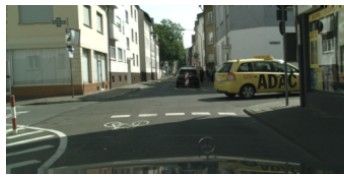 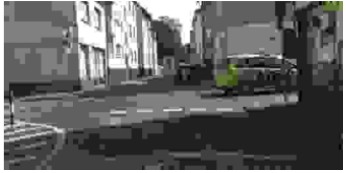 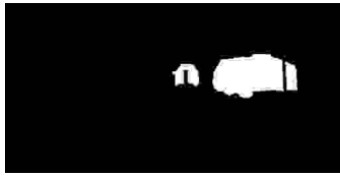

Original Image    Compressed Image    Compressed One-Hot Map

Figure 8: Example of how the JPEG compression affects the original image and a sample of the one-hot encoded maps. The compressed image loses informative content, while the one-hot encoded maps are minimally affected by the compression.

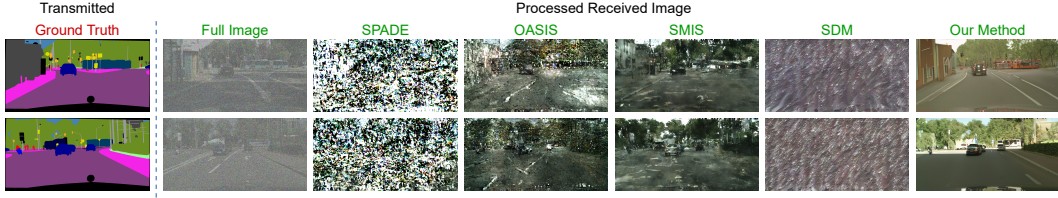

Figure 9: Other comparisons for transmitted semantics and a fixed PSNR value of 10.

for such a real-world problem. Thus, they are not robust to the heavy corruption the channel may implicate in transmitted information. This may result in noisy, unprecise, and corrupted generated content that can not be considered reliable from a communication perspective. From a quantitative point of view, we should consider that the signal-to-noise ratio of the transmitted maps with a channel PSNR$= 30$ is $84.6211$, while for a channel PSNR$= 10$ the signal-to-noise ratio of the transmitted content is equal to $1.6324$. This means that the content received by the diffusion model is extremely degraded.

To experimentally validate our claims, we perform a further comparison with ControlNet (Zhang & Agrawala, 2023), a recent powerful conditional method for diffusion models. We report sample images in Fig.7. When passing the information corrupted by the channel to ControlNet, the model is not able to extract a meaningful semantic map due to the noise added by the channel, especially with low PSNRs, i.e. PSNR$< 20$. Therefore, despite the high quality of generated images, the samples poorly preserve the semantic information that is the key to reliable communication systems. In addition, we test with ControlNet with any noise added to text information, which is an optimal condition for ControlNet and quite unrealistic since in a communication framework the textual caption should be extracted by the sender and transmitted over the channel that may add distortions.

## D  ADDITIONAL EXPERIMENTAL RESULTS

**Image vs Binary map transmission.** In order to validate the claims of the main paper regarding map compression, we show an example in Fig.8. The JPEG compression drastically reduces the informative content of the image. On the contrary, when applied to one-hot encoded binary maps, the content remains almost unchanged. For this reason, we can apply an aggressive compression algorithm to the transmitted maps in order to reduce the number of transmitted bits.

**Further comparisons.** Figure 9 provides additional comparisons for a given semantics under fixed channel conditions equal to PSNR $= 10$. While most of the samples generated from existing methods are heavily corrupted by noise, our method provides clearer images, while better preserving semantic features.

**Tests with different channel noises.** We run experiments simulating multiple channel noises other than AWGN. In detail, once we trained the model with AWGN, we then test the generalization ability of the proposed method with Poisson noise and a mixture of Poisson and Gaussian noises. We report the results for these channel noises with fixed PSNR=10 in Tab. 6. As it is clear for Tab. 6, the

Table 6: Results for different channel noises (Gaussian, Poisson, mixture) with fixed PSNR=10.

| Noise | Model | LPIPS↓ | mIoU↑ |
|---|---|---|---|
| AWGN | SMIS Zhu et al. (2020) | 0.603 | 0.901 |
| | Ours | **0.549** | **0.905** |
| Poisson | SMIS Zhu et al. (2020) | 0.631 | 0.795 |
| | Ours | **0.595** | **0.842** |
| Mixture | SMIS Zhu et al. (2020) | 0.639 | 0.822 |
| | Ours | **0.599** | **0.864** |

Table 7: Ablation study with fixed channel PSNR=10 for the proposed FDS block against a Swin-UNet (SUNet) denoising network Fan et al. (2022).

| Method | Params | FLOPS | Storage Memory | LPIPS↓ | mIoU↑ |
|---|---|---|---|---|---|
| FDS (ours) | 0M | 4G | 0.0GB | **0.549** | **0.905** |
| SUNet Fan et al. (2022) | 99M | 60G (+1400%) | 1.1GB | 0.575 | 0.869 |

proposed method is robust to different types of channel corruptions, preserving good performance across every experiment, as measured by both the LPIPS and mIoU metrics. Furthermore, these results highlight the generalizability of the proposed method that is robust to different kinds of noise even though it was trained in the AWGN scenario.

**Alation study for FDS block.** In order to evaluate the effectiveness of the proposed fast denoising semantic (FDS) block, we compare it with a Swin UNet Transformer (SUNet) Fan et al. (2022) for image denoising. Our proposed method has several advantages over a denoising network. First, it has no trainable parameters so it does not require to be trained when the scenario changes. Second, it has very light computations, therefore it does not affect the number of FLOPs of the model or the memory for checkpoints storage, as instead required by a denoising network. Table 7 shows the results of the proposed FDS module against the SUNet denoising network. The table confirms our intuition and the denoising model adds a consistent number of FLOPs to the computations, as well as more storage memory for saving the checkpoints to obtain similar results, actually worse than the FDS module.

**Downstream tasks metrics.** Although the downstream tasks are not the major scope of this paper, we provide additional evaluation metrics for the object detection and depth estimation tasks. We run DETR Carion et al. (2020) for object detection and DPT Ranftl et al. (2021) for depth estimation after we regenerate the images at the receiver side. Visual comparisons among methods and across different channel conditions are shown in Fig. 1 and Fig. 3. For metrics comparison, we select a challenging scenario where the channel noise is fixed at PSNR=10, and we compare our method with the best-performing comparison, that is SMIS Zhu et al. (2020). We report metrics results in Tab. 8, where we compute mAP and mAP50 for object detection and the RMSE for depth estimation. As Tab. 8 shows, the proposed method clearly outperforms the comparison method in terms of quantitative evaluation in both object detection and depth estimation downstream tasks.

**Generated samples from our model.** We report additional examples of our method generation. We randomly select three semantics and then report the generated samples from our method with seven different PSNR values. Results are shown in Fig. 10. Our method is able to generate good quality

Table 8: Quantitative metrics for downstream tasks (object detection and depth estimation).

| Task | Obj det. | | Depth est. |
|---|---|---|---|
| Model | mAP↑ | mAP50↑ | RMSE↓ |
| SMIS Zhu et al. (2020) | 0.230 | 0.451 | 44.102 |
| Ours | **0.390** | **0.666** | **14.530** |

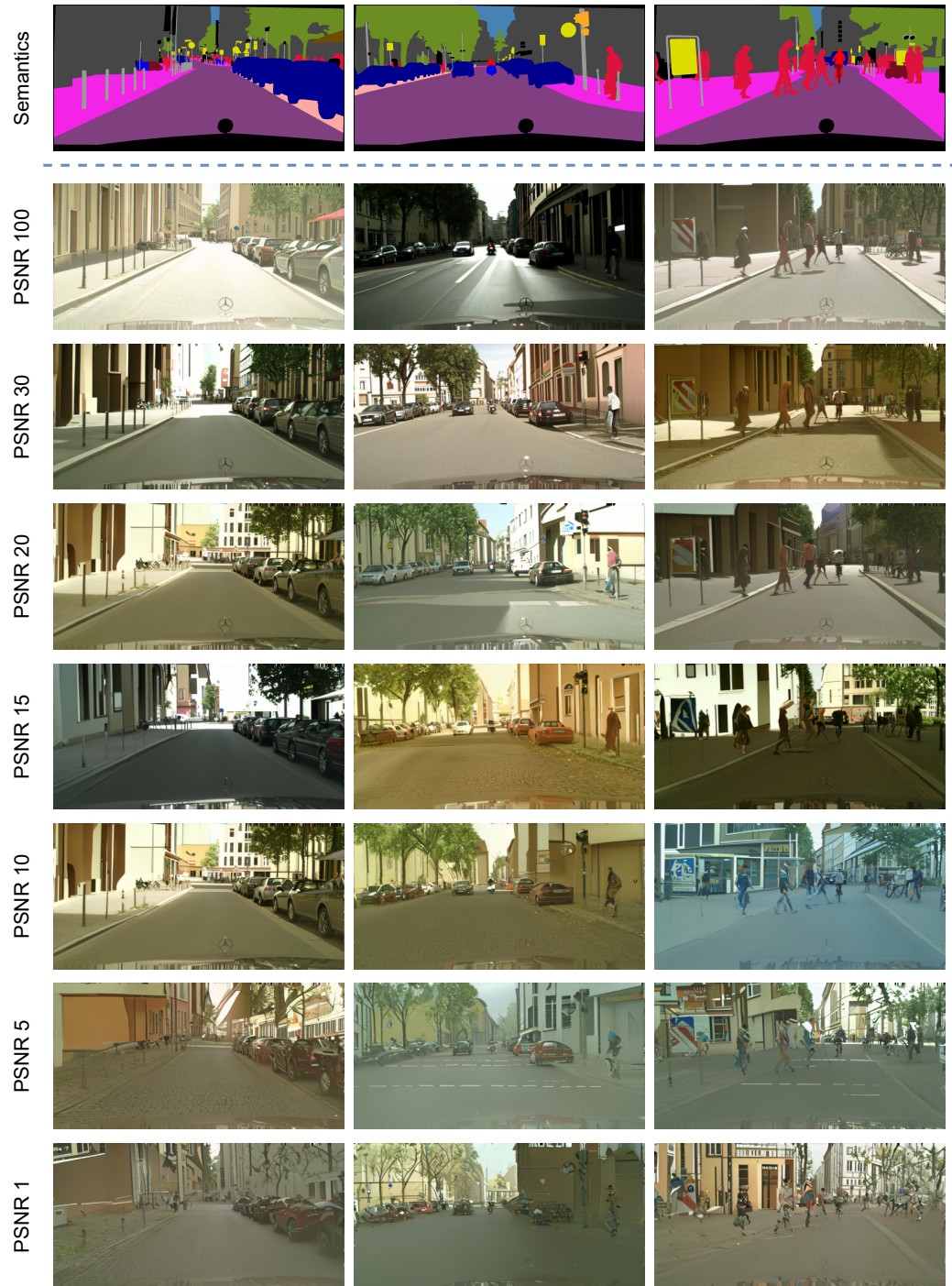

Figure 10: Generatd samples of our method from the transmitted semantics under different PSNR values for simulating various channel conditions.

images even in the case of extremely degraded channel conditions corresponding to PSNR values equal to 5 and 1. Indeed, objects and positions are still evident in these scenarios too.

