# OpenReview forum: "Generative Semantic Communication: Diffusion Models Beyond Bit Recovery"
_ICLR.cc/2024/Conference — Submitted to ICLR 2024_

### Official Review · Reviewer_4cUJ · 2023-10-31

**Soundness:** 3 good
**Presentation:** 2 fair
**Contribution:** 2 fair
**Rating:** 5
**Confidence:** 3

**Summary:**

This paper incorporates denoising diffusion probabilistic models (DDPM) to a new application, the semantic communication using deep generative models. The semantic communication process cares about preservation of semantic meanings instead of all details. The authors design a system utilizing DDPM as the generative model to recover the transmitted bits that contain semantic meanings of an image. The experiments show promising results on transmitted recovered images.

**Strengths:**

+ The paper contributes to a new application of generative model, which seems to be quite important in the communication field.
+ The authors thoroughly explain the architecture details in the paper
+ The result show performance improvements over baselines

**Weaknesses:**

There are several questions I'm hoping the authors can address:

- Although the generative model aims to be semantic preserving, the training method still uses recovering original image (where every pixel matters) as the objective function. This seems to be conflicting with the motivation.
- Pragmatic compression (preserving useful information in the bottleneck) [1] seems to be quite related. How would the authors compare with this line of work?
- In terms of writing, I feel it would be nice to have more overarching sentences explaining the model design instead of going into too details in the experiment section. There also seems to be a lack of explanation on the baselines in experiment section. This is especially important on less established benchmarks and tasks.
- There is no obvious modeling novelty on generative models or compression + diffusion algorithm.

Overall the paper is working on an interesting direction. But the paper needs some more work on justifying the training choice of generative model, making writing more clear and experiment section more informative.

[1] Pragmatic Image Compression for Human-in-the-Loop Decision-Making

**Questions:**

See above.

---

> ### Author Response · Authors · 2023-11-18
> **On the training method**
>
> We would like to thank the Reviewer for giving us the opportunity to clarify this point. The diffusion model is trained with such a loss function since it does not directly evaluate the quality of the reconstructed image, but it estimates the error between the true noise and the predicted noise by the network, as can be seen in Equation (8). Therefore, the network is trained to learn to estimate the noise so that it can be then removed during sampling, and to do so, a proper denoising loss is used.
>
> Moreover, in an ideal communication scenario, the goal of the receiver is always to recover the original signal as transmitted. However, communications involve many issues so it is not always possible to recover the original signals. Semantic communications aims at exploiting the idea of recovering the meaning of information rather than the signal itself that contains it. The objective is therefore to generate an image that preserves the semantics but, when possible, comes as close as possible to the original information.

---

> ### Author Response · Authors · 2023-11-18
> **On the comparison with pragmatic compression and additional explanations**
>
> We would like to thank the Reviewer for pointing out this comparison with the interesting field of pragmatic compression. In this paper, we propose a framework for semantic communications, this includes a semantic extractor, the transmission of the extracted semantic content that may be corrupted by the channel, and the generation of the images in the data space according to the received semantics. In this scenario, pragmatic compression can be seen as a special case of the proposed semantic communication framework, in which no corruption is applied to the features over the channel transmission, even though this is quite unrealistic in real-world communication systems. Therefore, the scope of our paper is not solely to semantically compress the information, but also to make the receiver robust and ensure that it can properly process such information even though it is corrupted from the transmission. Thus, we build a generative semantic communication framework that extracts the semantics and then it is able to leverage it to guide the generation process even in the case of heavy channel corruption.
>
> On the writing and additional explanations. We have revised the paper as suggested by the Reviewer, and we have added further explanations on the scenario and the baseline methods in Appendix A and B of the revised manuscript. We would like to thank once again the Reviewer for giving us the opportunity to clarify these aspects.

---

> ### Author Response · Authors · 2023-11-18
> **Further explanations on the novelty**
>
> We would like once again to thank the Reviewer for the possibility of clarifying this aspect. The scope of the paper is not proposing a novel diffusion algorithm, but rather an entirely novel generative framework for semantic communication, which is a new paradigm in communication. Our novelty relies on different aspects: First, we present a novel strategy to make the proposed semantic diffusion model robust to any noise corruption in the conditioning data, Our methodology opens the path to novel resilient generative models. Indeed, conditional generative models are usually tested under ideal conditions (no noise), while this is not always true, as in applications where the conditioning data may be corrupted (as an example, from a noisy channel in communications). Indeed, in such cases, known generative models fail to properly generate images, as we show in Figure 1, all over the paper, and remarkably also in Figure 7. Since we are proposing, and carefully evaluating, a novel framework, we start building it upon state-of-the-art diffusion model algorithms. In the literature, there exist several novel frameworks that involve existing diffusion models, that are often exploited as-they-are with pretrained checkpoints [1,2,3, and many others]. On the contrary, in this paper, we have rewritten the training procedure and we retrain the core diffusion model. We believe that the proposed framework contains a valuable contribution to the application of diffusion models to cases where the data used to condition the generative process is distorted, as well as providing, to the best of our knowledge, the first powerful generative framework for semantic communications. Secondly, we introduce a novel generative semantic communication framework, whose core is a semantic diffusion model at the receiver that receives corrupted semantic maps and applies fast denoising before involving them to guide the generation process of the diffusion model. To the best of our knowledge, this is the first semantic diffusion model trained and fully leveraged for the purpose of semantic communication in real-world scenarios.
> Thanks to the Reviewer’s suggestion, we have better highlighted the novel aspects in the revised version of the paper in order to clarify some important aspects of our work.
>
> [1] Liu, Haohe et al. “AudioLDM: Text-to-Audio Generation with Latent Diffusion Models”, ICML, 2023.
>
> [2] W. Hong, M. Ding, W. Zheng, X. Liu, and J. Tang, “CogVideo: Large-scale pretraining for text-to-video generation via transformers, ICLR, 2023.
>
> [3] A. Nichol, P. Dhariwal, A. Ramesh, P. Shyam, P. Mishkin, B. McGrew, I. Sutskever, and M. Chen, “GLIDE: Towards photorealistic image generation and editing with text-guided diffusion models”, ICML, 2021.

---

### Official Review · Reviewer_ckTn · 2023-11-01

**Soundness:** 2 fair
**Presentation:** 3 good
**Contribution:** 2 fair
**Rating:** 3
**Confidence:** 2

**Summary:**

This paper presents a framework for recovering images in semantic communication systems, by leveraging diffusion models. The semantic masks are sent, which are extracted from a semantic segmentation model. The mask may be altered because of noises during the communication. The images are synthesized using the semantic mask, according to a diffusion-based semantic image synthesis model. To further improve the synthesis performance, the proposed method first performs a fast denoising on the received semantic masks, which shows useful for better synthesis performance.

**Strengths:**

+ Overall, the idea to leverage a diffusion model to recover raw images to reduce communication costs sounds interesting.

+ The proposed method shows better synthesis results under the proposed setup, than several state-of-the-art semantic image synthesis models.

**Weaknesses:**

- The synthesis model is not novel. Diffusion models are popular nowadays. What are the new technical things in this work. Overall, the idea for fast denoising or training with noisy masks are not novel, which are straight solutions.

- In section 4.1, it is mentioned DETR is applied for evaluation. However, in the tables of experiments, mIoU is used to report the semantic similarity. In my understanding, mAP should be used for object detection and mIoU is for semantic segmentation.

- In section 4.1, depth estimation is mentioned for evaluation, however, which table does show the results of depth estimation? This is confusing for readers.

- What is the performance for clean semantic mask as the input? The core component is this work is a semantic image generator, therefore, it is important to show the proposed method is better than previous semantic image generation method.

**Questions:**

How to handle one-to-many mapping in image generation? For example, some low-level information may be missing with this solution, such as the colors of objects are incorrect, comparing the raw images and synthesized images.

---

> ### Author Response · Authors · 2023-11-17
> **On the object detection and depth estimation metrics**
>
> We would like to thank the Reviewer for giving us the opportunity to clarify these points that we understand may not be clear. The first and most important purpose of our evaluation is to evaluate if the proposed method is able to preserve the transmitted semantics, as we are proposing a generative semantic communication framework whose scope is to transmit semantic information and regenerate semantic-preserving content. To this end, we compute the mIoU metrics to compare the original semantic maps and the ones computed from the regenerated images. This is the crucial evaluation for a semantic communication framework.
> Then, as a further evaluation of the power of our generative semantic communication framework, we propose a comparison also considering specific tasks the receiver may have to solve, that may be object detection or depth estimation. Originally, we did not include any objective metrics in the paper, as we noticed that from the visual inspections, our method clearly outperformed other comparisons without needing objective comparisons. However, we agree with the Reviewer that specific metrics for these tasks could further clarify the advantages of our proposed strategies Hence,  we computed the metrics the Reviewer suggested and inserted them in Appendix D of the revised manuscript. Although these tasks are not the main problems the proposed method is aimed to solve, from this evaluation it is clear once again that our method performs better than alternative approaches.

---

> ### Author Response · Authors · 2023-11-17
> **On the comparison with clean maps and other semantic generation methods**
>
> The performance of the method with clean semantic maps as input is reported in Tables 1, 2, and 3 under the scenario in which the PSNR=100 (first column of the results), as it means that essentially no noise is applied to the transmitted content, therefore the semantic maps are clean.
> As the Reviewer correctly pointed out, the core of this work is the semantic generator at the receiver side. For this reason, we compared the proposed method with several other semantic image generative models that exist in the deep learning literature. Regarding semantic communication image generators, very few methods exist and none of them is tailored to the task of autonomous driving/remote control or involve these datasets, since they are rather simple methods and perform experiments on very simple datasets (CIFAR10, CelebA, …). Indeed, the proposed paper is extremely novel as instead it proposes a diffusion-based semantic communication framework that is able to work in real-world situations, such as complex datasets and heavily degraded channel scenarios.

---

> ### Author Response · Authors · 2023-11-17
> **On the one-to-many mapping**
>
> The advent of semantic communication has completely revolutionized communication purposes in some fields of application. While classical communication systems aim at exactly recovering the transmitted content, semantic communication frameworks have the scope of transmitting and recovering the semantics of the original image. Therefore, for the purposes of semantic communication, the style information, such as the colors, is not important information to recover as long as the receiver preserves the semantic information. In our case, at each generation the receiver preserves the key aspect of the transmitted information, that is the semantic map of the image, even though useless details such as generated colors may differ.

---

> ### Author Response · Authors · 2023-11-17
> **On the new technical things**
>
> We would like once again to thank the Reviewer for the possibility of clarifying this aspect. The scope of the paper is not proposing a novel diffusion algorithm, but rather an entirely novel generative framework for semantic communication, which is a new paradigm in communication. Our novelty relies on different aspects: First, we present a novel strategy to make the proposed semantic diffusion model robust to any noise corruption in the conditioning data, Our methodology opens the path to novel resilient generative models. Indeed, conditional generative models are usually tested under ideal conditions (no noise), while this is not always true, as in applications where the conditioning data may be corrupted (as an example, from a noisy channel in communications). Indeed, in such cases, known generative models fail to properly generate images, as we show in Figure 1, all over the paper, and remarkably also in Figure 7. Since we are proposing, and carefully evaluating, a novel framework, we start building it upon state-of-the-art diffusion model algorithms. In the literature, there exist several novel frameworks that involve existing diffusion models, that are often exploited as-they-are with pretrained checkpoints [1,2,3, and many others]. On the contrary, in this paper, we have rewritten the training procedure and we retrain the core diffusion model. We believe that the proposed framework contains a valuable contribution to the application of diffusion models to cases where the data used to condition the generative process is distorted, as well as providing, to the best of our knowledge, the first powerful generative framework for semantic communications. Secondly, we introduce a novel generative semantic communication framework, whose core is a semantic diffusion model at the receiver that receives corrupted semantic maps and applies fast denoising before involving them to guide the generation process of the diffusion model. To the best of our knowledge, this is the first semantic diffusion model trained and fully leveraged for the purpose of semantic communication in real-world scenarios.
> Thanks to the Reviewer’s suggestion, we have better highlighted the novel aspects in the revised version of the paper in order to clarify some important aspects of our work.
>
> [1] Liu, Haohe et al. “AudioLDM: Text-to-Audio Generation with Latent Diffusion Models”, ICML, 2023.
>
> [2] W. Hong, M. Ding, W. Zheng, X. Liu, and J. Tang, “CogVideo: Large-scale pretraining for text-to-video generation via transformers, ICLR, 2023.
>
> [3] A. Nichol, P. Dhariwal, A. Ramesh, P. Shyam, P. Mishkin, B. McGrew, I. Sutskever, and M. Chen, “GLIDE: Towards photorealistic image generation and editing with text-guided diffusion models”, ICML, 2021.

---

### Official Review · Reviewer_M1DV · 2023-11-01

**Soundness:** 3 good
**Presentation:** 3 good
**Contribution:** 3 good
**Rating:** 6
**Confidence:** 3

**Summary:**

This paper primarily addresses the noise issue in semantic communication and proposes a generative method based on the diffusion model for the recovery of transmitted bit sequences. The method is mainly divided into two parts, firstly, an FDS block is designed to remove the noise from the semantic mapping, and then the denoising ability of the diffusion model is utilized to train on the noisy data. Experimental evaluations are conducted on two datasets, Cityscapes and COCO-Stuff, and the experimental results show that the proposed method is advantageous in strong-noise scenarios and can substantially compress the transmitted content to improve communication efficiency.

**Strengths:**

1. The novel introduction of the diffusion model in semantic communication has contributed to the richness of this research area.
2. The proposed Fast Denoising Semantic Block (FDS) seems to be simple but effective for channel noise.
3. The experimental evaluation is rigorous by assessing the quality of the recovered images in terms of several metrics such as mIoU, LPIPS, FID, and depth estimation.
4. The experimental results are inspiring, especially in strong noise scenarios (PSNR<10). In addition, the binary bit transmission substantially improves communication efficiency.

**Weaknesses:**

1. There are deficiencies in the setting of noise conditions. In this paper, the authors only used white Gaussian noise of different intensities to review the method. However, other noises such as Poisson noise or a mixture of varying noises may occur during the actual transmission and the authors need to further evaluate the real-world relevance of the method.
2. The novelty of the paper is relatively weak, because, except for the FDS module, the diffusion model and the classifier-free guidance are already existing methods. The authors need to highlight the improvements made to these two components.
3. The structure of the semantic diffusion model appears to be complex and fine-grained, and the recovery process of the data may consume a lot of computational resources, and its feasibility in real-world applications needs to be further discussed.

**Questions:**

1. The classifier-free guidance should be added in ablation experiments to evaluate its effectiveness.
2. I would like to know the computational efficiency of the method, preferably in comparison with some typical generative and non-generative methods.
3. Since the FDC module seems to be simple and generic, could the authors combine it with other existing methods to validate the effectiveness of the module?

---

> ### Author Response · Authors · 2023-11-17
> **On the additional experiments with different noise scenarios**
>
> We would like to thank the Reviewer for her/his positive feedback and suggestions that can improve the quality and robustness of our work. Follow the Reviewer’s suggestions, we performed additional experiments with different types of noise, such as the Poisson noise and a mixture of Gaussian and Poisson noise. We report the results in Appendix D of the revised manuscript and a summary table here. As highlighted from the experiments, our method is very robust to different kinds of noise, having the ability to regenerate complex images even though the received semantic information is heavy corrupted by various kinds of noise (Gaussian, Poisson, and mixture). Moreover, even in these different noise scenarios, the proposed framework still outperforms comparisons according to all the metrics we computed.
>
> | Noise   | Model                                    | LPIPS$\downarrow$ | mIoU$\uparrow$ |
> |---------|------------------------------------------|-------------------|----------------|
> | AWGN    | SMIS        | 0.603             | 0.901          |
> |         | Ours                                     | **0.549**         | **0.905**      |
> | Poisson | SMIS        | 0.631             | 0.795          |
> |         | Ours                                     | **0.595**         | **0.842**      |
> | Mixture | SMIS         | 0.639             | 0.822          |
> |         | Ours                                     | **0.599**         | **0.864**      |

---

> ### Author Response · Authors · 2023-11-17
> **Highlight of the improvements**
>
> We would like once again to thank the Reviewer for the possibility of clarifying this aspect. The scope of the paper is not proposing a novel diffusion algorithm, but rather an entirely novel generative framework for semantic communication, which is a new paradigm in communication. Our novelty relies on different aspects: First, we present a novel strategy to make the proposed semantic diffusion model robust to any noise corruption in the conditioning data, Our methodology opens the path to novel resilient generative models. Indeed, conditional generative models are usually tested under ideal conditions (no noise), while this is not always true, as in applications where the conditioning data may be corrupted (as an example, from a noisy channel in communications). Indeed, in such cases, known generative models fail to properly generate images, as we show in Figure 1, all over the paper, and remarkably also in Figure 7. Since we are proposing, and carefully evaluating, a novel framework, we start building it upon state-of-the-art diffusion model algorithms. In the literature, there exist several novel frameworks that involve existing diffusion models, that are often exploited as-they-are with pretrained checkpoints [1,2,3, and many others]. On the contrary, in this paper, we have rewritten the training procedure and we retrain the core diffusion model. We believe that the proposed framework contains a valuable contribution to the application of diffusion models to cases where the data used to condition the generative process is distorted, as well as providing, to the best of our knowledge, the first powerful generative framework for semantic communications. Secondly, we introduce a novel generative semantic communication framework, whose core is a semantic diffusion model at the receiver that receives corrupted semantic maps and applies fast denoising before involving them to guide the generation process of the diffusion model. To the best of our knowledge, this is the first semantic diffusion model trained and fully leveraged for the purpose of semantic communication in real-world scenarios.
> Thanks to the Reviewer’s suggestion, we have better highlighted the novel aspects in the revised version of the paper in order to clarify some important aspects of our work.
>
> [1] Liu, Haohe et al. “AudioLDM: Text-to-Audio Generation with Latent Diffusion Models”, ICML, 2023.
>
> [2] W. Hong, M. Ding, W. Zheng, X. Liu, and J. Tang, “CogVideo: Large-scale pretraining for text-to-video generation via transformers, ICLR, 2023.
>
> [3] A. Nichol, P. Dhariwal, A. Ramesh, P. Shyam, P. Mishkin, B. McGrew, I. Sutskever, and M. Chen, “GLIDE: Towards photorealistic image generation and editing with text-guided diffusion models”, ICML, 2021.

---

> ### Author Response · Authors · 2023-11-17
> **On the computational resources and the the classifier-free guidance**
>
> **Computational resources.** We would like to thank the Reviewer for pointing out this aspect. As the Reviewer correctly wrote, the computational requirements are a delicate point. In this paper, our scope is introducing a very novel generative semantic communication framework to show the capabilities and potentials of properly designed diffusion models in semantic communication systems. Since nothing similar exists in the literature, we focus our attention on presenting the overall framework and the ability to regenerate contents with preserved semantics even in the case of heavy channel corruption. We did not focus on the computational efficiency part of the diffusion model that, instead,  needs a detailed and ad-hoc study. Indeed, for a communication system, several aspects should be considered, such as the time for sampling, the energy consumption, and the latency of the transmission. Therefore, just applying a fast inference or similar methods in this work would be belittling, since facing all these aspects to solve the computational curse would require specific studies that we plan to do in a future dedicated paper.
> However, to better highlight these aspects and the future works that should be done about them, we have inserted and discussed this point in the limitations part of the main corpus of the revised manuscript, and we would like to thank the Reviewer again for her/his insightful comments and suggestions.
>
> **Classifier free-guidance.** According to the Reviewer’s suggestion, we conducted an additional ablation study on the classifier-free guidance effectiveness. The results in the table below (fixed PSNR=10) prove once again the effectiveness of the classifier-free guidance that has been already widely tested [1,2,3].
>
> | Method                                | LPIPS | mIoU  |
> |---------------------------------------|-------|-------|
> | no classifier-free guidance | 0.631 | 0.802 |
> | with classifier-free guidance            | **0.549** | **0.905** |
>
> [1] H. Ni et al. “Conditional Image-to-Video Generation with Latent Flow Diffusion Models”, IEEE/CVF CVPR, 2023, pp. 18444-18455.
>
> [2] D. Lee et al. “Draft-and-Revise: Effective Image Generation with Contextual RQ-Transformer”, NeurIPS, 2022.
>
> [3] R. Rombach et al. “High-Resolution Image Synthesis with Latent Diffusion Models”, 2022 IEEE/CVF CVPR, 2021, pp. 10674-10685.

---

> ### Author Response · Authors · 2023-11-17
> **On the validation of the FDS module**
>
> We would like to thank the Reviewer for her/his suggestion. We developed the FDS module in order to be lightweight and to not require additional training, but, to the best of our knowledge, we are not aware of recent similar denoising blocks distinguished from denoising networks that perfectly work. On the contrary, instead of the FDS module, we could have inserted a more specific block, such as a powerful denoising network to remove noise from the transmitted maps. However, this network would require ad-hoc training for each dataset and consistently affect the computational resources, the FLOPs, and the memory at the receiver side. However, since the final scope of this paper is not to denoise the information but to involve such information to guide the generation process, we do not need perfect denoised maps, as long as they are clean enough to properly condition the diffusion model. Therefore, the FDS module performs a fast denoising that is sufficient to improve the quality of transmitted maps to guide the generation almost at cost zero.
> However, to further validate the effectiveness of the proposed FDS block, in addition to the ablation study of Subsection 4.3, we perform an additional ablation experiment comparing FDS with a Swin UNet transformer network [1] that performs denoising. We report results in Appendix D of the revised manuscript. As can be seen from the results, the proposed method has several advantages over the denoising network. First, it has no trainable parameters so it does not require to be trained when the scenario changes. Second, it has very light computations, therefore it does not affect the number of FLOPs of the model or the memory for checkpoints storage, as instead required by a denoising network. Moreover, the quantitative results confirm our intuition and the denoising model adds a consistent number of FLOPs to the computations, as well as more storage memory for saving the checkpoints to obtain similar results, actually slightly worse than the FDS module.
>
> | Method                 | Params | FLOPS           | Storage Memory | LPIPS$\downarrow$ | mIoU$\uparrow$ |
> |------------------------|--------|-----------------|-----------------|-------------------|----------------|
> | FDS (ours)             | 0M     | 4G              | 0.0GB           | **0.549**         | **0.905**      |
> | SUNet (Fan et al., 2022) | 99M    | 60G (+1400%) | 1.1GB           | 0.575             | 0.869          |
>
> [1] C.-M. Fan, T.-J. Liu, and K.-H. Liu, SUNet: swin transformer unet for image denoising, IEEE International Symposium on Circuits and Systems (ISCAS), pp. 2333–2337, 2022.

---

### Official Review · Reviewer_q6SF · 2023-11-03

**Soundness:** 2 fair
**Presentation:** 3 good
**Contribution:** 3 good
**Rating:** 5
**Confidence:** 4

**Summary:**

This work tries to consider semantic communication and visual generation at the same time with a new framework. This framework enables more robustness to corrupted conditioning in generation while preserving the transmitted layout as possible. It can also be viewed as a communication-friendly or corruption-robust layout generation framework.

**Strengths:**

Overall, I think the targeted issue of this paper, generative semantic communication, is very interesting and of high industry values. It seems that this new framework have high potentials of being applied to semantic compression or coding for machines.

The proposed framework is reasonable and clearly stated in this paper.

The experiments demonstrate the effectiveness of the proposed method on semantic segmentation.

**Weaknesses:**

Although this work has a very attractive starting point, it also has some obvious limitations：

1. It is not clear for the boundary/difference on the task settings between the target in this work and semantic compression. This work measures the fidelity of transmitted layout by comparing the accuracy of semantic segmentation under communication conditions with similar PSNR. However, it is puzzling why the transmission bit rate is not also one of the optimization targets of the model, like training a neural network based codec.

2. The work does not compare the performance of the proposed framework with directly using the transmitted layouts for semantic evaluation. It is unclear for the role of diffusion-based generative models from the perspective of communication.

3. It is somewhat overly simplistic for using the accuracy of semantic segmentation and the quality of generated images as the evaluation criteria for semantic communication. How about the optimization results for bit rates? And how about the effectiveness on other semantic downstream tasks?

**Questions:**

Please kindly see some detailed questions in the weakness part. I will adjust my final score based on author responses to my questions.

---

> ### Author Response · Authors · 2023-11-17
> **On the difference with semantic compression**
>
> We would like to thank the Reviewer for her/his positive feedback, insightful comments, and for giving us the opportunity to clarify these points.
>
> We would like to thank the Reviewer for giving us the opportunity to clarify this point. In this paper, we propose a framework for semantic communications, this includes a semantic extractor, the transmission of the extracted semantic content that may be corrupted by the channel, and the generation of the images in the data space according to the received semantics. Therefore, semantic compression can be seen as a special case of the proposed semantic communication framework, in which no corruption is applied to the semantics over the channel transmission, even though this is quite unrealistic in real-world communication systems. Therefore, the scope of our paper is not solely to semantically compress the information, but also to make the receiver robust and ensure that it can properly process such information even though it is corrupted from the transmission. Thus, we build a generative semantic communication framework that extracts the semantics and then it is able to leverage it to guide the generation process even in the case of heavy channel corruption.

---

> ### Author Response · Authors · 2023-11-17
> **On directly transmitting the layout**
>
> We would like to thank the Reviewer for the opportunity to better explain this point. There exist scenarios for which transmitting the layout only is sufficient for the task the receiver has to solve (for example, statically building the semantic landscape for machines only). However, there also exist scenarios in which it is desirable to regenerate the images that can provide more insightful information, such as remote control in which a human has to remotely control the device, or some scenarios of autonomous driving in which estimating the objects distance from regenerated images is easier than from the flat semantic layout. In this paper, we want to cover the last scenarios for which solely involving the layout is not sufficient, therefore, we focus our attention on these applications. In the revised version of this manuscript, we better clarified this scenario and our choices.

---

> ### Author Response · Authors · 2023-11-17
> **On the evaluation and bit rates**
>
> We would like to thank the Reviewer for giving us the opportunity to highlight this point of our work. We conduct a bit rate evaluation in Subsection 4.2. We compare the number of transmitted bits with respect to the classical communication scheme and, as it is clear from the results, our method is able to crucially reduce bandwidth usage and obtain better performance (92% reduction) under this comparison too. Thanks to the Reviewer’s suggestion, we have better highlighted this evaluation in the revised version of the manuscript.
>
> As the reviewer has correctly pointed out, the evaluation of semantic communication frameworks is still an open problem due to the fact that this is a very emerging topic. All the recent papers on the topic suggest that the evaluation of such framework should not rely on the bit rate accuracy, as the scope of the communication is not recovering the exact bitstream but rather the semantics of the transmission [1,2,3]. However, common metrics for semantic communication frameworks have not been introduced in the literature, and some works on image semantic communication propose to evaluate the recovered image with the LPIPS metrics [1,3,4]. Therefore, in this work, we followed this recipe and evaluated our framework with the LPIPS metrics first, but we also introduced a novel evaluation for a semantic communication framework. Indeed, as we are interested in preserving the semantics of the image, we evaluate this aspect by comparing the semantic maps involving the mIoU metrics. This is certainly a novel aspect in semantic communication frameworks and we believe this will be a practice to follow in the future. Finally, we also want to be sure that generated images are of realistic and of good quality and then evaluate them with the FID metrics.
>
> [1] J. Dai, P. Zhang, K. Niu, S. Wang, Z. Si, and X. Qin. Communication beyond transmitting bits: Semantics-guided source and channel coding. IEEE Wireless Communications 30 (4), 170-177, 2022.
>
> [2] X. Luo, H.-H. Chen, and Q. Guo. Semantic communications: Overview, open issues, and future research directions. IEEE Wireless Communication, 29(1):210–219, 2022.
>
> [3] S. Barbarossa, D. Comminiello, E. Grassucci, F. Pezone, S. Sardellitti, and P. Di Lorenzo. Semantic communications based on adaptive generative models and information bottleneck. IEEE Communication Magazine, 2023.
>
> [4] E. Agustsson, M. Tschannen, F. Mentzer, R. Timofte, and L. V. Gool, Generative Adversarial Networks for Extreme Learned Image Compression, IEEE/CVF International Conference on Computer Vision (ICCV), 221-231, 2019.

---

> ### Author Response · Authors · 2023-11-17
> **On other semantic downstream tasks**
>
> We would like to thank the Reviewer for giving us the opportunity to highlight this aspect. Together with the regeneration of the semantic content at the receiver side, we also perform some downstream tasks in our work. In detail, on the regenerated images, we perform experiments on object detection and depth estimation, in order to further validate the effectiveness of the proposed approach in a variety of scenarios. Examples of the obtained results can be found in Fig.1 and Fig.3, where we report the generated scenes with the bounding boxes obtained by DETR and the estimated depth from DPT. From these figures, it is clear how our method can effectively be used for several downstream applications, preserving the ability to recognize objects and estimate scene depth. To further validate these results, we additionally computed three metrics to have a quantitative evaluation of such downstream tasks, from which it is again clear that the proposed method outperforms comparisons. We added them in the Appendix D of the revised manuscript. We would like to thank once again the Reviewer for her/his insightful suggestions that can improve our work.

---

### Author Response · Authors · 2023-11-18
**Summary of changes**

We would like to thank the Reviewers for their positive comments and their insightful suggestions that can improve the quality of our work. Here is a summary of the changes we have made to the paper according to the Reviewers’ suggestions. Major changes are highlighted in blue in the revised manuscript.

1) We tested the performance of the proposed method under different noise types, proving its effectiveness and generalizability, according to all the objective metrics we computed, in the case of additive white Gaussian noise (AWGN), Poisson noise, and a mixture of noises.

2) We performed ablation studies for the FDS module, showing the advantages it brings in comparison with a powerful denoising network that consistently affects the computations at the receiver side without gaining performance advantages.

3) We better highlighted the differences between our work and the topics of semantic and pragmatic compression, which are special cases of our scenario in which the extracted semantic information is transmitted over the communication channel that can corrupt such information.

4) We have better highlighted the advantages of the proposed method on the bit rate where our model reduces the bandwidth usage by 92%, and added explanations on some limitations.

5) We added further explanations and details, and improved the overall writing quality, according to the Reviewers’ suggestions.

6) We computed objective metrics for downstream tasks in order to show that the proposed method objectively outperforms comparisons in these tasks too.

We believe that all the suggestions from the Reviewers helped improve the quality and the robustness of our work and we would like to thank them once again for their insightful comments.

---

### Author Response · Authors · 2023-11-21
**On the Authors-Reviewers discussion**

Dear Reviewers and Area Chair,

The deadline for the discussion phase is approaching and our rebuttal has not received any reply yet.

We would like to thank all the Reviewers for their comments and suggestions that helped to improve the quality and robustness of our work. We tried our best to address the raised concerns.

It would be great if you could parse our responses and let us know if there is anything that we can do to further improve the paper.

Thanks again for your effort.

The authors

---

### Meta-Review · Area_Chair_1CCD · 2023-12-11

**Metareview:**

This paper introduces a generative framework that combines diffusion models with semantic communication to recover important information from transmitted data.

The reviewers gave mixed feedback regarding the paper, with ratings of 6, 5, 5, 3.
 The reviewers appreciated the contribution of the paper to the interesting research problem, semantic communication, the paper studies, and the experimental evaluation on the effectiveness of the proposed method.
  However, there were concerns raised about the novelty of the synthesis model, the boundary between the target work and semantic compression, and the computational resources required for the proposed method.

While the authors addressed some of these concerns in their rebuttal by providing additional explanations and clarifications, the reviewers and AC believe the paper still needs further refinement and clarification, and suggest that the paper is not ready for acceptance at this time.

**Justification For Why Not Higher Score:**

The rejection is based on the consensus reached by reviewers.

**Justification For Why Not Lower Score:**

N/A

---

### Decision · Program_Chairs · 2024-01-16

Reject